# SLC38A10 Deficiency in Mice Affects Plasma Levels of Threonine and Histidine in Males but Not in Females: A Preliminary Characterization Study of SLC38A10^−/−^ Mice

**DOI:** 10.3390/genes14040835

**Published:** 2023-03-30

**Authors:** Frida A. Lindberg, Karin Nordenankar, Erica C. Forsberg, Robert Fredriksson

**Affiliations:** Molecular Neuropharmacology, Department of Pharmaceutical Biosciences, Uppsala University, 751 24 Uppsala, Swedenrobert.fredriksson@farmbio.uu.se (R.F.)

**Keywords:** SNAT10, knockout, amino acid homeostasis, gene expression analysis, solute carrier

## Abstract

Solute carriers belong to the biggest group of transporters in the human genome, but more knowledge is needed to fully understand their function and possible role as therapeutic targets. SLC38A10, a poorly characterized solute carrier, is preliminary characterized here. By using a knockout mouse model, we studied the biological effects of SLC38A10 deficiency in vivo. We performed a transcriptomic analysis of the whole brain and found seven differentially expressed genes in SLC38A10-deficient mice (*Gm48159*, *Nr4a1*, *Tuba1c*, *Lrrc56*, *mt-Tp*, *Hbb-bt* and *Snord116/9*). By measuring amino acids in plasma, we found lower levels of threonine and histidine in knockout males, whereas no amino acid levels were affected in females, suggesting that SLC38A10^−/−^ might affect sexes differently. Using RT-qPCR, we investigated the effect of SLC38A10 deficiency on mRNA expression of other SLC38 members, *Mtor* and *Rps6kb1* in the brain, liver, lung, muscle, and kidney, but no differences were found. Relative telomere length measurement was also taken, as a marker for cellular age, but no differences were found between the genotypes. We conclude that SLC38A10 might be important for keeping amino acid homeostasis in plasma, at least in males, but no major effects were seen on transcriptomic expression or telomere length in the whole brain.

## 1. Introduction

Solute carriers (SLC) are transporters consisting of 66 families (abbreviated as SLCXX, where XX is the number of the family; the full nomenclature of this group of transporters is described by [1]) and over 400 members have so far been discovered [2,3]. The SLCs are the biggest group of transporters in the human genome [1] and several of these genes have been associated with disease [4]. However, there is still a lack of research in the area of characterization of all members [5]. One family is the SLC38 group, which contains neutral amino acid transporters [6]. This family has 11 members, often referred to as SNATs (sodium-coupled neutral amino acid transporters), the preferred substrates of which are alanine and glutamine along with a few others that vary between members [6,7]. The latter members of the SLC38 family, SLC38A9-11, are evolutionarily older than others in the family, and SLC38A10, which is the focus of this paper, was present before the split of animals and plants [8].

The characteristics among the SLC38 members can be divided into system A- or N-type transporters, where A-type transport aliphatic amino acids, whereas N-type have a narrower substrate profile with glutamine, histidine and asparagine as the preferred substrates [7]. System A-type transport can be inhibited by 2-methylamino-isobutyric acid (MeAIB; through competition with the natural substrate of the transporter), and system N-type transport can also be driven by an antiport of H^+^ [6]. SLC38A10 has a bidirectional transport of L-alanine, L-glutamate, L-glutamine and D-aspartate, and has an efflux of L-serine, when measured in *Xenopus laevis* oocytes through overexpression transport assays [9]. The bidirectional transport of amino acids is a characterization of system N transporters, although SLC38A10 has been suggested to be classified as a system A transporter due to its transport of MeAIB [9].

In mammals, the SNATs vary in expression levels in different tissues, suggesting that they differ in function. A few of them have been suggested to be important for both the glutamine–glutamate cycle in the brain (SNAT1, 2, 3, 5 and possibly 6, 7 and 8) as well as for liver function, e.g., for urea production and the glucose–alanine cycle in the interorgan liver–muscle cycle (SNAT2, 3, 4 and 5 [6,10,11,12,13,14,15,16,17,18,19]). Amino acid transporters are also important in the placenta to transfer amino acids to the fetus (SNAT1, 2, 3 and 4 [7]). The expression of *Slc38a10* is present in most peripheral tissues and the brain, with a high expression in the pituitary, eye and lung [20], as well as expression through the gastrointestinal tract in rats [21]. In mice, *Slc38a10* was found abundantly expressed in the brain [9], and amino acid starvation upregulated *Slc38a10* in mouse primary cortex cells [22]. SLC38A10 is expressed in both astrocytes and neurons in the brain [9] and is localized to the Golgi and the endoplasmic reticulum [23]. Human SLC38A10 expression also points to ubiquitous expression throughout the body (Human Protein Atlas v. 22.0 [24]). However, the biological importance of *Slc38a10* is still unclear.

SNAT9 has been identified as a transceptor in the mechanistic target of the rapamycin complex 1 (mTORC1) pathway, localized in the lysosomal membrane and senses luminal arginine [25,26,27,28]. SNAT2 is also believed to function as a transceptor for mTOR activation, although localized in the plasma membrane [29,30]. In vitro knockdown of *Slc38a10* by siRNA resulted in a decreased protein translation [23], suggesting that SLC38A10 might function as a transceptor in a similar manner as SNAT2 and 9. The same research group also suggested that SLC38A10 might be important for cellular stress resistance and that an SLC38A10 deficiency might affect mitochondrial function [31]. The closely related SLC36 family is a family of transceptors [32], so it is possible that some of the SLC38 members, apart from SNAT2 and SNAT9, might function in a similar manner. 

The aim of the current study was to investigate the biological importance of SLC38A10 in vivo with the use of a knockout mouse model. The same knockout model has in a previous phenotyping screen been reported to have a lower body weight than a control [33,34]. Here, we focused on gene expression in the whole brain (transcriptomics), measured plasma levels of amino acids, thyroid hormones, and insulin, as well as the gene expression of other SNATs in selective tissues to make a preliminary characterization of SLC38A10-deficient mice. 

## 2. Materials and Methods

### 2.1. Animals

All animal work was approved by a local ethical committee (Uppsala) and followed Swedish legislation on animal welfare. Mice were kept in an animal facility with a controlled environment: 12 h light/dark cycle with lights on at 7 am, temperature at 20–24 °C, ventilation, humidity (45–65%) and daily supervision. Mice had *ad libitum* access to food (pellets R3 (21% protein, 52% carbohydrates and 5% fat), Lantmännen, Sweden) and water. 

A knockout first allele [35,36] was used to generate SLC38A10^−/−^ mice, which were bought from International Mouse Phenotyping Consortium (IMPC, www.mousephenotype.org, [33], accessed 1 September 2020). Heterozygous (B6Dnk;B6N-Slc38a10^tm2a(EUCOMM)Wtsi^/H) and control mice were bought from the International Mouse Phenotyping Consortium (IMPC, [37]) and kept for in-house breeding. The first heterozygous males from the breeding were in turn bred with C57BL/6J females (Taconic M&B, Ejby, Denmark). Further breeding was carried out from in-house bred offspring. 

In this paper, SLC38A10^−/−^ mice will be abbreviated as KO, SLC38A10^+/−^ as HET and SLC38A10^+/+^ as WT. Individual weights were measured at postnatal weeks 5 and 6 for WT, HET and KO mice of both sexes (males: 25 WT, 29 HET and 22 KO; females: 45 WT, 28 HET and 23 KO). 

### 2.2. Genotyping

Ear biopsies were obtained after weaning (3–4 weeks of age) and incubated in 75 µL of 25 mM NaOH and 200 µM EDTA at 95 °C for 40 min. After a cooling time of 5 min on ice, 75 µL of 40 mM Tris-HCl was added and the samples were then ready for PCR analysis. For the PCR, primers for the LacZ domain in the knockout construct (as described in [34]), and primers for the wildtype gene were designed. 

### 2.3. β-Galactosidase Staining

One mouse from each genotype (WT, HET and KO) and sex was euthanized through cervical dislocation, and the brains were dissected and divided into coronal sections by using a brain matrix. Sections were stored in Pierce™ IP lysis buffer and Pierce™ protease and phosphatase inhibitor mini tablets (Thermo Fisher Scientific, Uppsala, Sweden) at 4 °C for 2 h and thereafter used for β-galactosidase staining. The β-galactosidase reporter gene staining kit (#GALS, Sigma-Aldrich (St. Louis, MO, USA)) was used to verify the knockout construct (Figure 1A). Briefly, the sections were washed twice in 1 mL of PBS, fixed in 1 mL of fixation solution for 25 min, washed twice with PBS and incubated at 37 °C for 2 h with 1 mL of staining solution. Image acquisition of the brain sections was performed with a Leica M125 microscope with a ProgRes C14 plus camera (Jenoptik) and the ProgRes CapturePro 2.8 Jenoptik Optical system. 

### 2.4. Blood Plasma Collection

Blood collection was conducted during the same two-hour time window for all mice (10–12 am, light phase). Mice were sedated with CO_2_ (50% medical air, 50% CO_2_). Thereafter, blood was collected through heart puncture with an EDTA (0.5 M)-coated syringe (Terumo, 1 mL) and needle (Sterican 27 G, 0.4 × 20 mm), and the mice were killed by cervical dislocation. The blood was transferred to 1.5 mL microcentrifuge tubes (Sarstedt) coated with 0.5 M EDTA, and additional EDTA was added to the blood (2 µL EDTA/1 mL of blood). The blood was thereafter centrifuged for 10 min at 3000 rpm at 4 °C. Plasma was collected and transferred to another tube, which was stored at −80 °C until further use. 

#### 2.4.1. Hormone Measurement in Plasma

Thyrotropin (TSH) and free thyroxine (fT4) were measured with a sandwich ELISA kit from MyBiosource (San Diego, CA, USA, #MBS777023 and #MBS3805923) and insulin with a sandwich ELISA from Millipore (Burlington, MA, USA, #EZRMI-13K). Plasma was thawed on ice and used according to the manufacturer’s protocol (except for incubation at 37 °C which was extended to 70 min for the TSH and fT4 assays, and the last incubation for the insulin assay was extended to 30 min). The standard curve for each assay was used to calculate the hormone concentration in each sample. Plasma from both male and female mice was used (7 WT and 7 KO per sex for TSH and fT4, 8 WT and 8 KO for insulin), as well as one pooled HET (from 7 HET mice) plasma sample from each sex. Animals were 24–26 weeks of age. All samples, blanks and standard curves were run in duplicate. Samples outside the standard curve (1 sample in insulin ELISA), or samples where the coefficient of variation between the duplicates was too big (>15%; 3 samples in fT4 ELISA) were excluded from the analysis. Assays were analyzed with the four-parameter logistic curve online data analysis tool provided by MyAssays Ltd. (http://www.myassays.com/four-parameter-logistic-curve.assay, accessed on 18 January 2022). One sample in the insulin ELISA was identified as an outlier according to Grubbs’ outlier test (GraphPad QuickCalcs, Boston, MA, USA, http://www.graphpad.com/quickcalcs/grubbs1 (accessed on 18 January 2022)) and was therefore removed from the analysis, although it is plotted in the graph. Analysis was conducted between WT and KO mice, for each sex separately, with an unpaired t-test with Bonferroni correction for multiple comparisons. 

#### 2.4.2. Amino Acid Measurements

Plasma was transported to the Swedish Metabolomics Centre (Umeå, Sweden), where an amino acid analysis of the plasma was performed. Only WT and KO mice (n = 8/genotype and sex) were included in this study. Males were 10–12 weeks of age, whereas females were between 10–20 weeks of age. 

##### Standards and Calibration Curve

Amino acid standards (alanine, arginine, aspartic acid, cysteine, glutamic acid, glycine, histidine, isoleucine, leucine, lysine, methionine, phenylalanine, proline, serine, threonine, tyrosine, valine, glutamine, asparagine, γ-aminobutyric acid (GABA), citrulline, ornithine, taurine, tryptophan, 5-HTP and kynurenine) were purchased from Sigma-Aldrich (St. Louis, MO, USA). Isotopically labeled amino acid standards (alanine (^13^C_3_, ^15^N), arginine (^13^C_6_, ^15^N_4_), aspartic acid (^13^C_4_, ^15^N), cystine (^13^C_6_, ^15^N_2_), glutamic acid (^13^C_5_, ^15^N), glycine (^13^C_2_, ^15^N), histidine (^13^C_6_, ^15^N_3_), isoleucine (^13^C_6_, ^15^N), leucine (^13^C_6_, ^15^N), lysine (^13^C_6_, ^15^N_2_), methionine (^13^C_5_, ^15^N), phenylalanine (^13^C_9_, ^15^N), proline (^13^C_5_, ^15^N), serine (^13^C_3_, ^15^N), threonine (^13^C_4_, ^15^N), tyrosine (^13^C_9_, ^15^N), valine (^13^C_5_, ^15^N), Citrulline (d4), GABA (^13^C_4_), glutamine (^13^C_5_), asparagine (^13^C_4_), ornithine (d6), tryptophan (d8), kynurenine (d4)) were obtained from Cambridge Isotope Laboratories (Andover, MA, USA). Stock solutions of each compound were prepared at a concentration of 500 ng/μL and stored at −80 °C. A 10-point calibration curve (0.01–100 pmol/µL) was prepared by serial dilutions and spiked with internal standards at a final concentration of 5 pmol/μL. Mass spectrometry grade formic acid was purchased from Sigma-Aldrich (St. Louis, MO, USA) and HPLC grade acetonitrile from Fisher Scientific (Fair Lawn, NJ, USA). 

##### Amino Acid Extraction

Amino acids in plasma were extracted by mixing 100 μL plasma samples with 300 μL of 90:10 (*v*/*v*) methanol/water solution containing IS at 2.5 nmol/μL. Each sample was extracted for 30 s using a mixer mill, incubated in the freezer for 1 h and centrifuged at 4 °C, 14,000 RPM, for 10 min. The supernatant was collected and stored at −80 °C until analysis.

##### Amino Acid Derivatization with AccQ-Tag

Extracted samples were derivatized by AccQ-Tag™ (Waters, Milford, MA, USA) according to the manufacturer’s instructions. Briefly, 10 μL of the extract was added to 70 μL of AccQ•Tag Ultra Borate buffer and finally, 20 μL of the freshly prepared AccQ•Tag derivatization solution was added, and the sample was immediately vortexed for 30 s.

Samples were kept at room temperature for 30 min followed by 10 min at 55 °C. For each batch, quality control samples and procedure blanks were included.

Calibration curves were prepared in a similar way by adding 10 µL of each point of the curve to 70 µL of AccQ•Tag Ultra Borate buffer and 20 μL of the freshly prepared AccQ•Tag derivatization solution. 

##### Amino Acids Quantification by LC-ESI-MSMS

The derivatized solutions were analyzed using a 1290 Infinitely system from Agilent Technologies (Waldbronn, Germany), consisting of a G4220A binary pump, a G1316C thermostatic column compartment and a G4226A autosampler with a G1330B autosampler thermostat coupled with an Agilent 6460 triple quadrupole mass spectrometer equipped with a jet stream electrospray source operating in positive ion mode.

Separation was achieved by injecting 1 μL of each sample onto a BEH C_18_ 2.1 × 100 mm, 1.7 μm column (Waters, Milford, MA, USA) held at 50 °C in a column oven. The gradient eluents used were H_2_O 0.1% formic acid (A) and acetonitrile 0.1% formic acid (B) with a flow rate of 500 μL/min. The initial conditions consisted of 0% B, and the following gradient was used with linear increments: 0.54–3.50 min (0.1–9.1% B), 3.50–7.0 (9.1–17.0% B), 7.0–8.0 (17.0–19.70% B), 8.0–8.5 (19.7% B), 8.5–9.0 (19.7–21.2% B), 9.0–10.0 (21.2–59.6% B), 10.0–11.0 (59.6–95.0% B), 11.0–11.5 (95.0% B) and 11.5–15.0 (0% B). From 13.0 min to 14.8 min, the flow rate was set at 800 μL/min for a faster equilibration of the column.

The MS parameters were optimized for each compound as described in the Appendix A. MRM transitions for the derivatized amino acids were optimized using MassHunter MS Optimizer software (Agilent Technologies Inc., Santa Clara, CA, USA). The fragmentor voltage was set at 380 V, the cell accelerator voltage at 7 V and the collision energies from 14 to 45 V. Nitrogen was used as the collision gas. The jet-stream gas temperature was 290 °C with a gas flow of 11 L/min, sheath gas temperature of 325 °C and sheath gas flow of 12 L/min. The nebulizer pressure was set to 20 psi and the capillary voltage was set at 4 kV. The QqQ was run in Dynamic MRM Mode with 2 min retention time windows and 500 msec cycle scans.

The data were quantified using MassHunter™ Quantitation software B08.00 (Agilent Technologies Inc., Santa Clara, CA, USA), and the amount of each amino acid was calculated based on the calibration curves. Grubbs’ outlier test was used on the results, and significant outliers were removed from the analysis but are plotted in the graphs (Figure 2C,F). An analysis was performed between the genotypes for each sex separately with unpaired *t*-tests with Bonferroni correction for multiple comparisons. A principal component analysis (PCA) was conducted to see any groupings between genotypes based on amino acid levels in blood plasma. The analysis was performed using the PCA command from the factoextra library in R version 4.2.0, with the scale.unit variable set to TRUE. The graphs were plotted using the fviz_pca_ind, fviz_pca_var and fvis_eig commands, also from the factoextra package.

#### 2.4.3. Olink Proteomics

Plasma samples from adult males (3 WT, 3 KO; 10–12 weeks of age) were used in the mouse exploratory panel (Olink Proteomics, Uppsala, Sweden), which is a panel that assays 92 proteins involved in a wide range of biological functions and pathways. Results are reported as Normalized Protein eXpression (NPX) values which are on a log2 scale. In total, 78 proteins were detected in the samples (minimum of two samples per group), analyzed with unpaired t-tests to compare WT and KO and were false discovery rate (FDR)-corrected with an FDR set to 0.05. 

### 2.5. RNA Sequencing

Brains from 10-week-old male mice (4 WT, 4 KO) were collected and stored at −80 °C until RNA extraction. Total RNA from the brain was extracted with the Absolutely RNA Miniprep Kit (Agilent Technologies). Briefly, tissues were homogenized in a Bullet blender (Next advance, Raymertown, NY, USA) in lysis buffer and β-mercaptoethanol. A pre-filter spin cup was used to filter the homogenate in a centrifuge at 4 °C at maximum speed. To precipitate the RNA, a 1:1 ratio of 70% ethanol (Solveco) was mixed with the filtrate, which thereafter was centrifuged through an RNA-binding spin cup. After DNase treatment, salt buffers were used to wash the filter before RNA elution. The RNA was stored at −80 °C until mRNA isolation.

The RNA was treated with the Ribo-Zero™ Magnetic Kit to remove ribosomal RNA and purified using Agencourt^®®^ RNAClean^®®^ XP Kit (Beckman Coulter, Brea, CA, USA). The RNA was then treated with RNase III according to the Ion Total RNA-Seq protocol (Life Technologies, Carlsbad, CA, USA) and purified with a Magnetic Bead Cleanup Module (Life Technologies). The size and quantity of RNA fragments were assessed on the Agilent 2100 Bioanalyzer system (RNA 6000 Pico kit, Agilent, Santa Clara, CA, USA) before proceeding to library preparation, using the Ion Total RNA-Seq Kit v2 (Life Technologies). The libraries were amplified according to the protocol and purified with a Magnetic Bead Cleanup Module (Life Technologies). The samples were then quantified using the Agilent 2100 Bioanalyzer system (High Sensitivity DNA kit, Agilent) and pooled followed by emulsion PCR and chip loading on the Ion Chef system using the Ion PI Hi-Q Chef Kit (Life Technologies). The samples were loaded on an Ion P1v3 Chip (4 samples per chip) and sequenced on the Ion Proton™ System using Ion PI™ Hi-Q™ (200 bp read length, Life Technologies) chemistry. 

The reads were mapped against the Ensemble (http://www.ensembl.org, accessed on 18 January 2022) *Mus musculus* genome assembly “*Mus_musculus.GRCm38.DNA.toplevel*” using STAR mapper [38]. The mapping was performed against a genome index generated with the Ensemble GTF file “*Mus_musculus.GRCm38.89.abinitio*” to direct mapping towards annotated and predicted genes. A total of 166,593,399 (average: 20,824,174 ± 8,067,525 per sample) transcripts were mapped uniquely and were used in subsequent analysis. The assembled transcripts were used in CuffLinks and Cuffmerge to obtain a final transcriptome assembly. Subsequently, Cuffquant and Cuffdiff were used to calculate differential expression, and, finally, the package CummRBound was used in R [39] to plot the results. The version for all CuffLinks tools was 2.2.1 [40,41,42,43].

### 2.6. Tissue Collection, Preparation and RNA Extraction

Adult male and female mice (approximately 11 weeks old, n = 4–6 per genotype and sex) were killed through cervical dislocation and tissues were collected. Directly after dissection, the tissues were put in RNAlater (Sigma-Aldrich) and put on dry ice. The tissues were stored at −80 °C until RNA extraction. 

RNA from the brain, kidney, liver and lung were extracted with the Absolutely RNA Miniprep Kit (Agilent Technologies, Santa Clara, CA, USA). Briefly, the tissues were homogenized in a Bullet blender (Next advance, Troy, NY USA) in lysis buffer and β-mercaptoethanol. A pre-filter spin cup was used to filter the homogenate in a centrifuge at 4 °C at maximum speed. To precipitate the RNA, a 1:1 ratio of 70% ethanol (Solveco) was mixed with the filtrate, which thereafter was centrifuged through an RNA-binding spin cup. After DNase treatment, salt buffers were used to wash the filter before RNA elution. RNA from skeletal muscle was extracted with the RNeasy Fibrous tissue mini kit (Qiagen, Hilden, Germany), by homogenization in a Bullet blender (Next advance, USA) in Buffer RPE and β-mercaptoethanol. The samples were thereafter handled according to the manufacturer’s protocol, including Proteinase K and DNase treatment before the elution of the RNA. An ND-1000 spectrophotometer (NanoDrop Technologies, Boston, MA, USA) was utilized to measure RNA concentration. RNA was stored at −80 °C until further use.

### 2.7. cDNA Synthesis and Quantitative Real-Time PCR

cDNA was conducted using the High-Capacity RNA-to-cDNA Kit (Applied Biosystems, MA, USA). For real-time PCR, 20 ng of template was used per reaction (assuming 1:1 transcription from RNA to cDNA), as well as 0.05 µL of each primer (100 µM), 1 µL DMSO, 0.5 µL 10× SYBRGreen (1:50,000, diluted in TE buffer pH 7.8, (Invitrogen, Waltham, MA, USA)), 0.2 µL dNTPs (25 mM, Thermo Scientific, Waltham, MA, USA), 0.08 µL Dream Taq, 3.6 µL Dream Taq buffer (Thermo Scientific) and water to obtain a reaction volume of 20 µL. After an initial denaturation of 30 s at 95 °C, 46 cycles of denaturation (95 °C) for 10 s, 30 s at primer annealing temperature and a 30 s elongation step (72 °C) were carried out before a final elongation step (72 °C) for 5 min and a melt curve analysis ranging from 55 °C for 81 cycles (10 s intervals of 0.5 °C). CFX Maestro (Bio-Rad, Sweden) software was used for the initial analysis of each run, and qbase+ software (version 3.2, Biogazelle, Zwijnaarde, Belgium—https://www.qbaseplus.com/, accessed on 25 May 2022) was used for further analysis, using multiple reference genes according to [44]. The reference genes used were *Actb*, *Tubb4b* and *H3c* for the brain samples, *Actb* and *Tubb4b* for the kidney, liver and lung samples, and *Tubb4b* and *H3c* for the muscle samples. Normalized data were logarithm transformed, analyzed using a one-way-ANOVA and corrected for multiple testing in qbase+. Primer efficiencies were calculated via LinRegPCR software, and the primers used are summarized in Table 1. Gene expression is presented as log2 fold differences compared to the WT group.

### 2.8. Relative Telomere Length Measurement

To measure the relative telomere length for the different genotypes, adult male mice (n = 5/genotype, 10–12 weeks of age) were euthanized through cervical dislocation, and the brains were dissected and directly put on dry ice. The brains were put in −80 °C until gDNA extraction was performed. gDNA was extracted with the DNeasy Blood and Tissue kit (Qiagen) according to the manufacturer’s protocol. Briefly, 20 ng of brain tissue (parts of cerebellum and cortex) was lysed with proteinase K overnight, followed by treatment according to protocol, before the samples were put on DNeasy spin columns, washed and thereafter eluted. The samples were diluted to 2 ng/µL and thereafter used in the Relative Mouse Telomere Length Quantification qPCR Assay Kit (Sciencell research laboratories, Carlsbad, CA, USA). A total reaction volume of 20 µL contained 2 µL gDNA, 2 µL of the primer set (Telomere primer set or Single Copy Reference (SCR) primer set), 3.6 µL DreamTaq buffer, 0.25 µL Hot-Start DreamTaq, 1 µL DMSO and 0.5 µL 10× SYBRGreen (1:50,000, diluted in TE buffer pH 7.8 (Invitrogen)) and dNTPs (25 mM, Thermo Scientific). The program was run according to the manufacturer: an initial denaturation at 95 °C for 10 min, 40 cycles of 95 °C (20 s), 52 °C (20 s) and 72 °C (45 s), and with a melt curve analysis at the end of the run. Ct values were obtained through the CFX Maestro software (Bio-Rad, Sweden). The SCR primer set was used to normalize the data, and differences were calculated through the Comparative ΔΔCq method. 

### 2.9. Proteomic Measurement in Brain

#### 2.9.1. Sample Preparation

Brains from seven-week-old males were dissected and kept at −80 °C until further use (n = 5/genotype, only KO and WT genotypes used). The brain tissue samples were thereafter homogenized in liquid nitrogen. Aliquots of 25 mg brain samples were homogenized in the presence of 1 mL of lysis buffer (1% β-octyl-glucoside, 8 M urea, protease inhibitor cocktail in PBS) for 60 s with a sonication probe (VibraCell 600 W, 1.5 cm probe) and for 30 min using ice in an ultrasound bath. A protease inhibitor cocktail (5 μL) was added during the sample preparation to prevent protein degradation. After homogenization, the samples were incubated for 90 min at 4 °C during mild agitation. The tissue lysates were clarified by centrifugation for 15 min at 16,000× *g* at 4 °C. Thereafter the total protein concentration in the samples was measured using the DC Protein Assay with bovine serum albumin (BSA) as standard. Aliquots corresponding to 30 μg protein were taken out for digestion. 

The proteins were reduced, alkylated and on-filter digested by trypsin using 3 kDa centrifugal filters (Millipore, Arklow, Ireland) according to a standard operating procedure. The collected peptide filtrate was vacuum centrifuged to dryness using a speedvac system. Dried peptides were resolved in 100 μL of 0.1% FA prior to nano-LC-MS/MS. 

#### 2.9.2. LC-MS/MS

The resulting peptides were separated in reversed-phase on a C18-column, applying a 150 min-long gradient, and electrosprayed on-line to a QEx-Orbitrap mass spectrometer (Thermo Finnigan). Tandem mass spectrometry was performed applying HCD. 

#### 2.9.3. Qualitative Analysis

Database searches were performed using the quantification software MaxQuant 1.5.1.2. The search was performed towards proteins from *M. musculus* proteome extracted from Uniprot, released in October 2016. The search parameters were set to Taxonomy: *M. musculus*, Enzyme: Trypsin. Fixed modification was Carbamidomethyl (C), and variable modifications were Oxidation (M) and Deamidated (NQ). The search criteria for protein identification were set to at least two matching peptides with a 95% confidence level per protein.

#### 2.9.4. Quantitative Analysis

Ten samples were included in the quantitative analysis. The ten RAW data files were quantitatively analyzed by the quantification software MaxQuant 1.5.1.2. Protein identification was performed by a search against the same in-house FASTA database for the qualitative analysis. A decoy search database including common contaminations and a reverse database was used to estimate the identification false discovery rate (FDR). The results of all the samples were combined into a total label-free intensity analysis for each sample. A two-tailed Student’s *t*-test was performed on the generated full protein expression list and a *p*-value < 0.05 was considered statistically significant. A ratio cut-off value was set to a two-fold change for upregulated proteins and a cut-off value of 0.5 for downregulated proteins. 

## 3. Results

We used a knockout-first allele (illustrated by [34]) in a mouse model to study the effect of SLC38A10 deficiency on the blood plasma levels of amino acids, thyroid-related hormones and insulin, as well as differentially expressed genes in the brain, liver, lung, muscle and kidney tissues in WT, HET and KO mice. 

### 3.1. Analysis of β-Galactosidase and Slc38a10 Expression, as Well as Weight in Young Mice

The promotor activity of the KO construct was confirmed with reporter gene staining, namely, the staining of the product of the *LacZ* gene, β-galactosidase. In Figure 1A, the results clearly demonstrate the lack of staining in a WT brain, whereas the most staining was seen in the KO brain as expected. Similar results were seen in both male and female mice (Figure 1A). To further verify the KO, RT-qPCR was used to confirm the lack of *Slc38a10* expression in KO mice. However, low levels of transcripts could be detected in four out of five tissues (Figure 1B). In the brain, kidney, lung and muscle, low levels of *Slc38a10* expression were detected in KO mice, whereas around half of the expression of WT mice was seen in HET mice. Expression in the muscle was less knocked down and was more spread within the KO group than in other tissues (Figure 1B). In the liver, no expression could be detected in KO mice, whereas the expression in HET mice was around the same fold change as in other tissues. The differences were statistically different in all tissues between all genotypes.

The body weight of the animals was measured, and the weights were found to be lower in the KO mice than in the HET and WT mice, for both males and females at both time points (Figure 1C). This agrees with earlier studies [33,34,37], indicating the robustness of this phenotype. 

### 3.2. Blood Plasma Levels of Amino Acids

The amino acid levels in the plasma were measured in adult mice of both sexes by LC-ESI-MSMS (Figure 2A–F). Of the 26 investigated amino acids, 3 were detected close to baseline which makes them comparable between groups, but the concentrations cannot be used as absolute concentrations (Figure 2B,E). In males, two amino acids were found to be at lower levels in KO mice compared to WT, namely, the essential amino acids threonine and histidine (Figure 2C), whereas in females, no differences between the KO and WT mice were detected (Figure 2D–F). In the female data, three mice in the WT group and one mouse in the KO group were older than the other animals in the respective group (16 or 20 weeks compared to 10 weeks). These data points are therefore marked with a blue border color in order to visualize the data completely since age is known to affect amino acid levels [45]. These data points did not have any major impact on the statistical analysis, which made us keep the data as it is, especially since these values were not continuously lower than other values for all amino acids. However, it incorporates an element of uncertainty in the data.

The amino acid data were later used in a PCA to form a scatter plot of each individual mouse (Figure 3). In this analysis, the male and female mice were separated into different halves of the plot, indicating that there is a sex difference regarding amino acid levels in blood plasma. In males, one can distinguish a grouping between KO and WT mice, where WT mice tend to localize more to the upper right quadrant and in the upper part of the vertical axis whereas KO males tend to localize more to the upper left quadrant and closer to the origin of the plot. In females, grouping between the WT and KO mice was not as clear. The WT females were mostly localized to the lower left quadrant and along the lower vertical axis of the plot, whereas the KO females were spread in both quadrants on different sides of WT females. The scree plot and loading plot that complement the scatter plot from the PCA can be seen in Appendix A.

### 3.3. Olink Proteomics and Metabolic Hormone Measurements in Plasma

Out of the 92 proteins in the Olink exploratory panel, 78 were detected in the plasma from adult WT and KO males. Initial analysis found seven proteins at lower levels in KO males (Figure 4A), namely: Il23r (Interleukin-23 receptor), Tnr (Tenascin-R), Dll1 (Delta-like protein 1), Plxna4 (Plexin-A4), Wisp1 (WNT1-inducible-signaling pathway protein 1), Rgma (Repulsive guidance molecule A) and Dlk1 (Protein delta homolog 1). However, after FDR correction, the differences were no longer of statistical significance. 

In six-month-old mice, the plasma levels of TSH, fT4 and insulin were measured (Figure 4B–D). In male KO mice, an increased level of TSH was observed compared to their WT counterparts, but no such effect could be seen in the females (Figure 4B). No statistical differences were seen for the insulin or fT4 levels (Figure 4C,D). 

### 3.4. Transcriptomic Data from Whole Brain

The RNA sequencing was carried out mainly as an exploratory experiment to seek an explanation as to why the phenotypic effects of SLC38A10 deficiency previously reported were surprisingly small [34]. The mRNA sequencing on brain tissue was performed on tissue from adult male mice and generated eight statistically differentially expressed genes between WT and KO mice (Table 2). The top one, *Slc38a10*, verifies the results from RT-qPCR (Figure 1C): the transcript could be found but at low levels, suggesting a knockdown of the protein rather than a complete knockout. Other genes that had an altered expression in the KO brains were *Gm48159*, *Nr4a1*, *Tuba1c*, *Lrrc56*, *mt-Tp*, *Hbb-bt* and *Snord116/9*. Note that the fold change in Table 2 is in WT compared to KO. 

### 3.5. Gene Expression Analysis in Brain, Kidney, Liver, Lung, and Kidney, as Well as Telomere Length Measurement in Brain

To discover the differential expression of other SNATs as possible compensation mechanisms consequently to SLC38A10 KO, we performed RT-qPCR on brain, liver, lung, muscle, and kidney tissue. *Mtor* and *Rps6kb1* were also examined since a correlation between SLC38A10 and the mTOR pathway seems to be present [23,46]. 

*Slc38a1-5* (Appendix A), *Slc38a7*, *Mtor*, and *Rps6kb1* (Appendix A) were found and analyzed for all tissues, whereas *Slc38a9* and *Slc38a11* (Appendix A) were not analyzed in the liver due to poor PCR efficiency and technical difficulties. The *Slc38a8* transcript was only efficiently amplified in the brain and muscle, hence they were the only tissues analyzed (Appendix A). However, none of the genes examined were differentially expressed in SLC38A10-deficient mice. It is possible that there are sex-specific effects for some of the genes, see Appendix A, for example, but due to the small group sizes, the analyses performed with sex collapsed. One should keep this in mind when observing the data, although no trends of a possible genotype effect could be observed when visually inspecting the data.

We hypothesized that a changed response to stress in neuronal cells from KO mice [31] could have an effect on neuronal health and increase their rate of aging. We used telomere length as an indicator of cell age and found that the relative telomere length did not differ between the genotypes (Appendix A). Although, a big spread within the HET and KO group and the relatively low number of animals used makes an interpretation of these results hard. As far as these results show, the telomere length was not affected by SLC38A10 deficiency. 

### 3.6. Brain Proteomics

The proteomic study was designed to detect any changes in the regulation of other SLCs, or changes in related pathways, and to possibly detect changes not seen on the transcriptomic level. The protein concentration and the number of identified proteins in the respective samples can be seen in Appendix A. However, no proteins detected in the LC-MS/MS fulfilled the cut-off ratio criteria. All proteins that were statistically significant between KO and WT brains are listed in Appendix A along with the ratio of respective proteins. One protein (Nipsnap3b) was uniquely identified in all 5 samples of group “KO” but never identified in any of the samples from group “WT”. Briefly, 14 proteins (Sf3a2; Vipas39; Etl4/Skt; Nfasc; Eml2; Cd9; Sec24b; Camk2n1/Camk2n2; Hmgcs1; Fam114a2; Tf; Acot1; Nhp2l1; Pllp) were identified in at least 3 out of 5 samples from group “KO” but never identified in any of the samples from group “WT”. No proteins were identified in all five samples of group “WT” but never identified in all samples of group “KO”. Nine proteins (Stt3b; Dgkh; Zc3h6; Mccc2; Mt1; Plcl1; Bola2; Pcdh10; Acsl3) were identified in at least three out of five samples from group “WT” but never identified in any of samples from group “KO”. 

## 4. Discussion

In the current study, an SLC38A10 knockout mouse was studied to understand the biological role of this amino acid transporter. The plasma levels of amino acids, proteomic exploratory assays and plasma hormones (insulin, TSH, and fT4) were measured. Gene expression analyses of the brain, kidney, liver, lung, and muscle as well as a relative telomeric measurement of the brain were also performed. 

### 4.1. Plasma Levels of Amino Acids Is Affected by SLC38A10

The effect of SLC38A10 deficiency on plasma levels of amino acids differed between male and female mice. In males, two of the measured amino acids differed between the genotypes, threonine and histidine, whereas no amino acids differed between WT and KO in female mice (Figure 2A–F). Threonine is an essential amino acid involved in lipid metabolism, protein synthesis and intestinal function (reviewed by [47]), and plasma levels have been demonstrated to correlate with the organ levels of threonine in rats [48]. When catabolized, threonine is oxidized to glycine and acetyl-CoA, and levels of glycine and threonine are positively correlated in the brain [48], which could possibly affect brain function if altered. Histidine is also an essential amino acid, important for the binding of metal ions in enzymes, as well as having antioxidant activity and buffering properties [49]. Histidine is also the precursor of histamine, which is important for immune function, appetite and cognitive function and more [49]. Decreased levels of these two amino acids could therefore affect a wide range of processes, which would be interesting to study further. Although just two amino acids were found to differ in males, a grouping based on genotype in the PCA could be seen (Figure 3), indicating that the amino acid profile is slightly changed in KO males compared to WT males.

Amino acid metabolism is known to be influenced by sex hormones [50,51,52], and from the data generated herein, a distinction between male and female amino acid profiles was detected (Figure 3). An SLC38A10 deficiency seems to affect the sexes differently as seen in the lack of differences between the genotypes in female mice in Figure 2 and the lack of genotype grouping in Figure 3. Age is another factor that is known to affect amino acid levels in blood [45,52], which makes the age spread among the tested female mice inconvenient (10–20 weeks). However, the removal of the three older females in the WT group and the one older female in the KO group did not affect the results from the analysis. This and the fact that these animals did not have continuously lower levels of all amino acids made us keep these data points in the analysis.

Another factor that will affect the amino acid levels in plasma is, of course, when mice had their last food intake. In this study, the mice were not fasted before blood sampling occurred, which is a drawback. We think that limiting the blood collection to a two-hour time period in the forenoon (during the light phase) partly controls for that. Since mice are nocturnal animals and do most of their eating during the dark phase, it is likely that they did not consume any food just before the blood collection. However, since this was not controlled in this case, we cannot rule out that feeding might have affected these results. 

Amino acid homeostasis is a complex process, and it is controlled by amino acid transporters with slightly different functions in the body. These transporters usually have one main function in the cell: a net transport of amino acids (a loader), or an exchange of amino acids (an exchanger) (reviewed by [53]). Commonly, SNATs are known to have a net transport of amino acids [6], but it is also possible that SLC38A10 has more of a transceptor function rather than a classical transporter function [23], which then could cause an imbalance in amino acid sensing when absent. Consequences could be altered plasma levels of amino acids, possibly with a greater effect in males than females. 

### 4.2. Effects on Plasma Levels of Insulin, TSH and fT4 in SLC38A10-Deficient Mice

Due to the high expression of *Slc38a10* in the pituitary gland [20] and previous reports of SLC38A10 KO mice having a decreased bone mineralization [54], we were interested in investigating if the SLC38A10 KO had an effect on thyroid function. Thyroid hormones (TH) are important in skeletal development, which has been described in several studies where different parts of the TH signaling system were knocked out in mice that gave rise to phenotypes of decreased bone mineralization (reviewed by [55]). However, the results from our study failed to show any drastic effect on TSH and fT4 plasma levels. The TSH levels in the male KO mice were found to be higher than in the WT males (Figure 4C), but the biological effect of this small increase is likely not of importance, which the non-affected levels of fT4 in these mice suggest. In humans, this could be a sign of subclinical hypothyroidism (reviewed by [56]) and could indicate a desensitized thyroid which could mean that problems might occur in the future.

### 4.3. Whole-Brain Transcriptomics

RNA sequencing was performed on the whole brain of adult male mice and generated eight transcripts that were statistically significantly changed. The *Slc38a10* transcript was the most affected transcript, which verifies the RT-qPCR results of the same gene (Figure 1C). Other transcripts affected were *Gm48159, Nr4a1, Tuba1c, Lrrc56, mt-Tp, Hbb-bt* and *Snord 116/9* (Table 2). It is worth noting that if the transcriptomic analysis had been conducted on specific brain regions instead of the whole brain, additional differences could have been discovered. It is difficult to interpret the transcriptomic data due to the same reason—the brain is a complex organ with different areas specialized in different functions. The altered transcripts found could mean different things in different brain regions, or in different cell types. However, these transcripts that were found to be differentially expressed indicate that these were greatly affected in the whole brain. 

Out of these transcripts, three were found to be of particular interest (*Nr4a1*, *Tuba1c* and *mt-Tp*). These are involved in processes regulating cell cycle regulation, proliferation and p53 expression. *Nr4a1* has been reported to be involved in processes such as apoptosis, carcinogenesis, cell cycle regulation and metabolism, as well as in the hypothalamo-pituitary–adrenal (HPA) axis, hypothalamo–pituitary–gonadal (HPG) axis and hypothalamo–pituitary–thyroid (HPT) axis [57]. *Tuba1c* encodes a subtype of α-tubulin that is a part of the microtubule, the cytoskeleton, and is involved in various cellular functions such as cell division, molecular transport, p53-signaling, glycolysis and glucogenesis [58,59]. *mt-Tp* is a mitochondrial t-RNA gene that is upregulated in the brains from KO mice, which could suggest an effect on protein synthesis in the mitochondria. Tripathi, Aggarwal and Fredriksson [31] reported that primary embryonic cortex cells deficient in SLC38A10 had altered p53 levels and mitochondrial function, suggesting that SLC38A10 might be important for cellular stress resistance. The above-discussed transcripts could very well be a part of this effect. Tripathi, Aggarwal and Fredriksson [31] also induced oxidative stress in the same cells and reported on a reduced stress response in those cells. *Hbb-bt* encodes the adult t-chain of hemoglobin and is involved in oxygen transport, hydrogen peroxide catabolic process and cellular oxidant detoxification [60]. These results imply a potential role of SLC38A10 in redox processes.

The remaining transcripts are more difficult to associate with SLC38A10. The transcript of *Lrrc56* is required for the assembly of dynein arms [61], but the function in the brain is unknown. *Snord 116/9* is a small nucleolar RNA (snoRNA), part of a snoRNA cluster of non-protein-coding RNAs that are involved in rRNA modification [62]. One study found that a deletion of the *Snord116* cluster lead to growth deficiency and hyperphagia in mice [63]; another study suggested that the same cluster may be involved in the regulation of epigenic factor expression that is important for circadian rhythm [64]. However, from just one differentially expressed snoRNA from a complete cluster, further conclusions cannot be drawn.

*Gm48159* is a novel antisense transcript to *Rspo3* (an activator of the Wnt signaling pathway), which suggests it to be an expression regulator of this gene. Since it is unknown how this transcript affects *Rspo3*, it is impossible to speculate about its effects. It is interesting, though, that this transcript is just as affected as the *Slc38a10* gene (Table 2), but the biological effect of it is unclear. 

### 4.4. Whole-Brain Proteomics

It is interesting to note that the protein of the *Nipsnap3b* gene was uniquely identified in all KO brains but in neither of the WT brains. However, the exact role of the NIPSNAP3B protein is not yet known, although it has a putative role in vesicular trafficking [65]. The 14 proteins that were only found in the KO brains (in at least three out of five samples) are involved in pre-mRNA splicing, vesicular trafficking, cell adhesion, metabolism and cytoskeletal structure [66]. The nine proteins found in only WT brains (in at least three out of five samples) are associated with the glycosylation, synthesis and metabolism of fatty acids, cell adhesion, GABA-signaling and ion-binding [66]. Together these results imply that metabolism and cell function could be affected in the brains from KO mice when compared to WT mice, which previously have been reported for primary cortex cells from the same mouse model [46]. 

### 4.5. General Discussion/Limitations of the Current Study

Based on the results obtained herein, amino acid levels in plasma, transcriptomics and proteomics are affected in male SLC38A10-deficient mice, but no compensatory expression of other SNATs was detected. However, similar results have not been obtained for females. Since RNA sequencing, proteomics and telomere length measurements were not performed on female mice, we cannot make the same conclusion for both sexes. Since no differences were seen in amino acid levels in the plasma in females, it is possible that we have sex-specific effects of this KO model, and by not implementing the female sex for these experiments we may have failed to detect potential effects on these parameters.

## 5. Conclusions

Based on the results from this study, the knockout-first allele was not efficient enough to achieve a global knockout of *Slc38a10* in all tissues, but rather a knockdown in all tissues examined, except for the liver where no transcripts were found and a knockout was achieved. However, this SLC38A10 knockdown causes, as previously reported, a lower body weight than in WT mice but also decreased plasma levels of the essential amino acids threonine and histidine in KO males. No effects were seen on the mRNA expression of other Slc38 members in the tissues examined (brain, liver, lung, muscle, and kidney), but transcriptomic analysis in the brain revealed a few transcripts involved in cellular stress and redox processes in males. 

Together with the fact regarding the evolutionary age of the latter SLC38 transporters, the mild effects seen in this study, and the fact that *Slc38a10* is widely expressed might indicate a more general involvement in maintaining the homeostasis of cells. If this homeostasis would be disturbed, it would most likely cause some sort of disturbance in vivo. However, the KO strategy is complicated, and compensation mechanisms are common with this approach [67].

## Figures and Tables

**Figure 1 genes-14-00835-f001:**
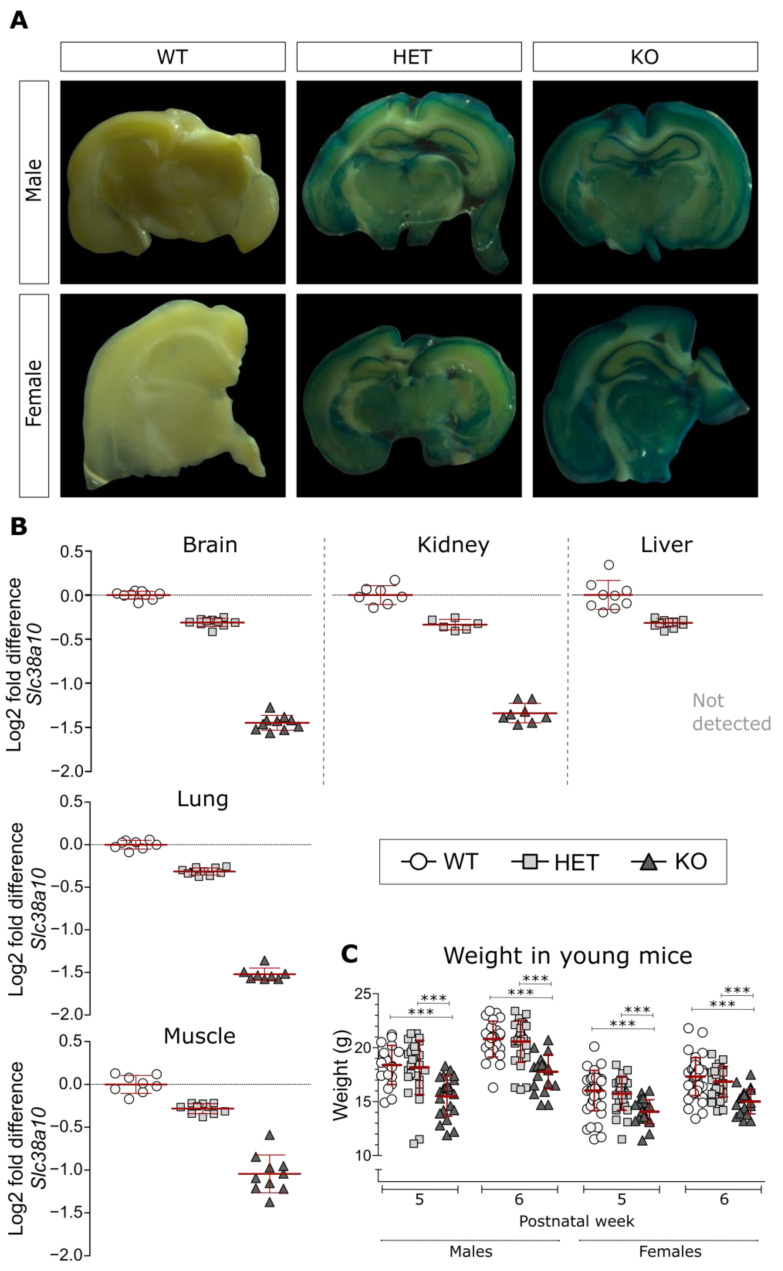
Display of β-galactosidase expression, *Slc38a10* gene expression and weight in SLC38A10-deficient mice. (**A**) Indirect staining of the β-galactosidase protein (encoded by the *LacZ* gene), showing the promotor activity in the genetically modified male and female mice, respectively, as well as lack of staining in the WT control. (**B**) *Slc38a10* relative expression in brain, kidney, liver, lung, and muscle from WT, HET and KO mice were measured by RT-qPCR using two to three reference genes. Expression is presented as a log2 fold change compared to the mean of the WT group. Analysis was conducted with one-way ANOVA with correction for multiple testing (according to qbase+ software version 3.2, Biogazelle, Zwijnaarde, Belgium—https://www.qbaseplus.com, accessed 25 May 2022). (**C**) Weight differences between young WT (n = 25 males; 45 females), HET (n = 29 males; 28 females) and KO (n = 22 males; 23 females) mice. Analysis was conducted with a two-way ANOVA (repeated measures) with Bonferroni correction, *** = *p* < 0.001.

**Figure 2 genes-14-00835-f002:**
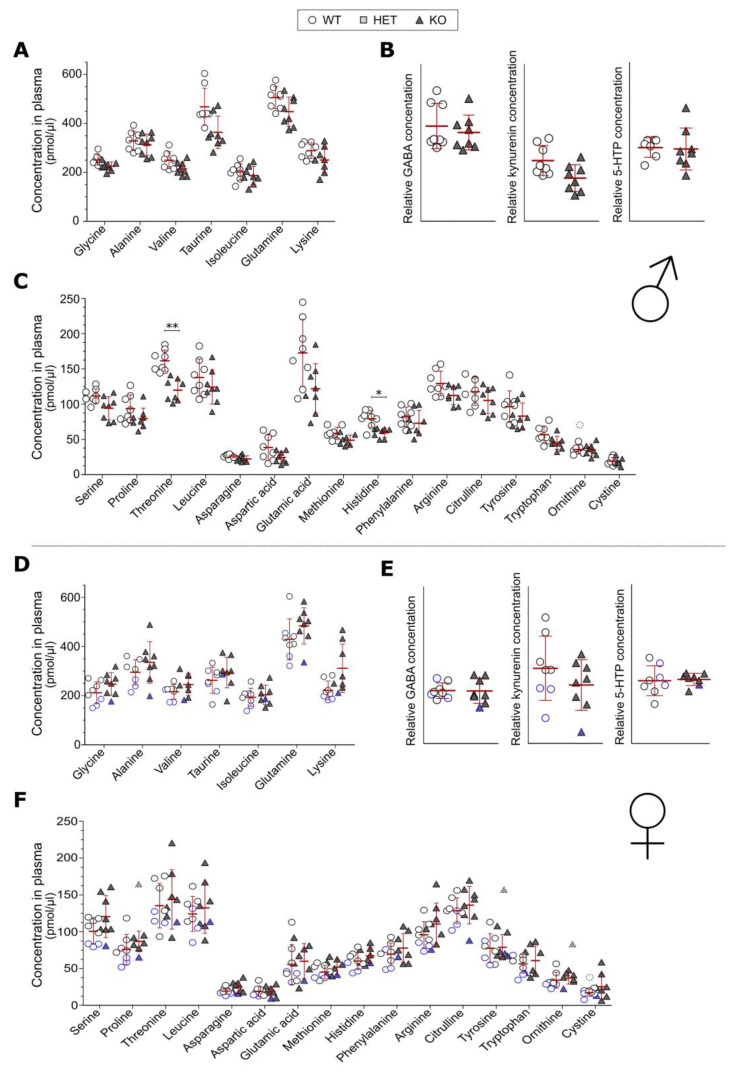
Amino acid levels in the plasma of SLC38A10 KO mice. Blood was collected from adult mice and the plasma was used for amino acid quantification. Amino acid levels were determined by LC-ESI-MSMS. Both (**A**–**C**) male (n = 7–8 WT, 8 KO) and (**D**–**F**) female (n = 7–8 per genotype) mice were used. Amino acids detected close to baseline are shown without absolute concentrations (**B**,**E**). Amino acid data were analyzed with unpaired t-tests with Bonferroni correction (adjusted *p*-value = 0.00197). Outliers (significantly different from the other data points according to Grubbs’ outlier test) were excluded from the analysis but plotted in the figure with dashed lines. Statistical differences between KO and WT mice are indicated with an asterisk sign (* *p* < 0.00197 and ** *p* < 0.0001).

**Figure 3 genes-14-00835-f003:**
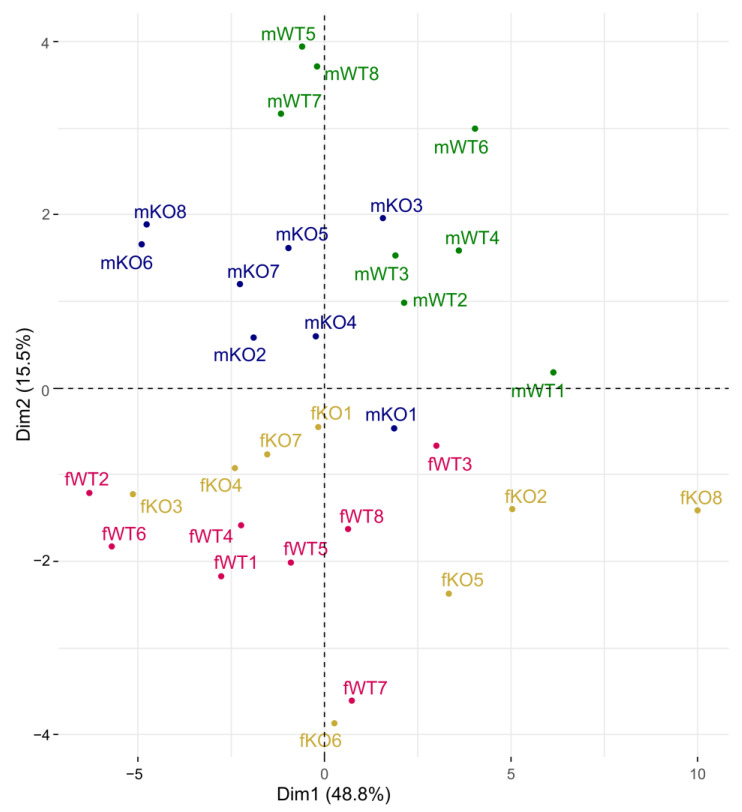
Score plot showing individual animals in a principal component analysis (PCA), where genotype and plasma levels of amino acids were of interest. Blood was collected from adult mice and the plasma was used for amino acid quantification. Amino acid levels were determined by LC-ESI-MSMS and used in a PCA to visualize groupings between sex and genotype. fKO, female knockout mouse; fWT, female wildtype mouse; mKO, male knockout mouse; mWT, male wildtype mouse. Numbers separating individuals within each grouping.

**Figure 4 genes-14-00835-f004:**
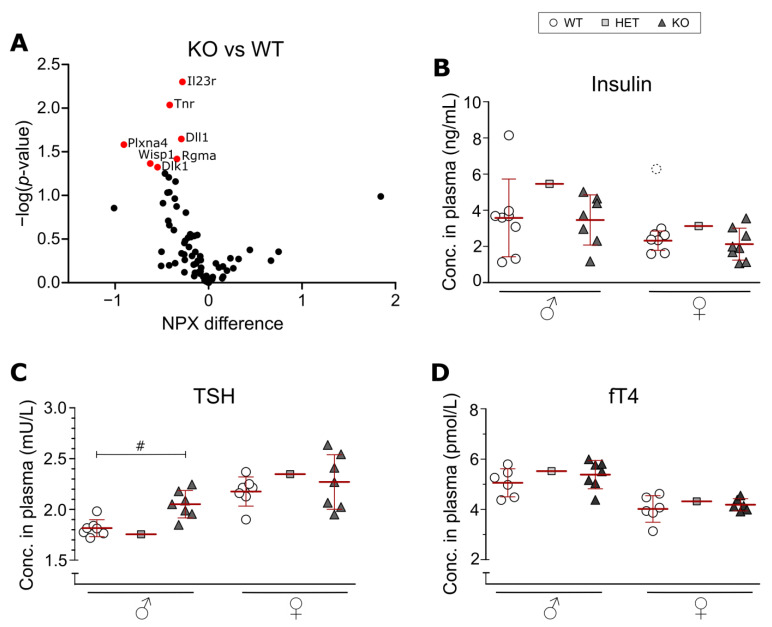
Protein expression levels in plasma of SLC38A10 KO mice. Blood was collected from adult mice, and the plasma was used for an Olink mouse exploratory panel and ELISA measurements of TSH, fT4 and insulin. (**A**) The mouse exploratory panel (Olink Proteomics AB, Uppsala, Sweden) was used to analyze 92 proteins related to a wide range of functions. In total, 78 proteins were detected, and each plotted with its NPX difference (KO relative to WT) against *p*-value. Data were analyzed with unpaired t-tests and with FDR correction (FDR set to 0.05). Proteins in red were statistically different between the genotypes after t-tests, but not after FDR correction. (**B**) Insulin (n = 8 WT, 7 KO males; 7 WT, 8 KO females), (**C**) TSH (n = 7 WT and 7 KO per sex) and (**D**) fT4 (n = 6 WT, 7 KO males; 6 WT, 6 KO females) levels in plasma were measured with sandwich ELISA. One pooled HET sample (from 7 animals) per sex was included in the experiment, but not statistically analyzed. Outliers (significantly different from the other data points according to Grubbs’ outlier test) were excluded from the analysis but plotted in the figure with dashed lines. KO and WT data were analyzed with an unpaired t-test with Bonferroni correction (adjusted *p*-value, # *p* < 0.01667). (**B**–**D**) Data are illustrated as individual data points and with mean ± standard deviation. NPX: Normalized Protein eXpression.

**Table 1 genes-14-00835-t001:** Primer sequences used in RT-qPCR.

Gene:	Primer (5′-3′):
*H3c*	Forward: ccttgtgggtctgtttga; Reverse: cagttggatgtccttggg
*Actb*	Forward: ccttcttgggtatggaatcctgtg; Reverse: cagcactgtgttggcatagagg
*Tubb4b*	Forward: agtgctcctcttctacag; Reverse: tatctccgtggtaagtgc
*mTOR*	Forward: acatttgaagaagcagag; Reverse: tgatctcctccatctctt
*Rps6kb1*	Forward: actagtgtgaacagagggcc; Reverse: ttcctccagaatgttccgct
*Slc38a1*	Forward: tcacgagagtcacgcagagatg; Reverse: gagcagtatgagaacaagcgaagc
*Slc38a2*	Forward: tctactcgctggttcttc; Reverse: aataaacttgtcacttccctta
*Slc38a3*	Forward: atatttgggatcattggt; Reverse: catgattcggaagtagaa
*Slc38a4*	Forward: agaacatcagacaaagac; Reverse: ttagtgttggtgtagagt
*Slc38a5*	Forward: tggaggtgtctggtctctaataac; Reverse: ggcagtgaggcaactctaagg
*Slc38a7*	Forward: tagccattgcggtctata; Reverse: gctccttcgacatcacag
*Slc38a8*	Forward: cgtggtgactcgggacag; Reverse: ctacaagccagggacactaagg
*Slc38a9*	Forward: tgctgtcattgctgtaat; Reverse: actgagaagaaccatcct
*Slc38a10*	Forward: tggtgaaggctccgaagaaagg; Reverse: acttggcttgggtctgaactgg
*Slc38a11*	Forward: actttcaattcggaacct; Reverse: catcagtgctaatcttgtg

**Table 2 genes-14-00835-t002:** Results from mRNA sequencing of the whole brain. Brains from adult male mice (4 WT, 4 KO) were dissected and RNA was extracted and used for mRNA sequencing. The mRNA sequences that were significantly altered in SLC38A10-deficient mice are shown in the table. Note that analysis is performed with KO set as control; hence, the log2 fold change is in WT brains compared to KO. Red color signifies an upregulation and green color a downregulation.

	log2	Adjusted	Gene
	Fold Change	*p*-Value	
ENSMUSG00000061306	4.261195835	4.92 × 10^-210^	*Slc38a10*
ENSMUSG00000110831	4.475565312	6.32 × 10^-9^	*Gm48159*
ENSMUSG00000023034	0.84989094	0.000108	*Nr4a1*
ENSMUSG00000043091	−0.70844378	0.004278	*Tba1c*
ENSMUSG00000038637	−1.25011731	0.006453	*Lrrc56*
ENSMUSG00000064372	−1.42276608	0.007218	*mt-Tp*
ENSMUSG00000073940	−0.77138591	0.008367	*Hbb-bt*
ENSMUSG00000096634	−1.36720258	0.009128	*Snord116/9*

## Data Availability

RNA-seq data are deposited at Array Express (E-MTAB-12561). The mass spectrometry proteomics data have been deposited to the ProteomeXchange Consortium via the PRIDE [68] partner repository with the dataset identifier PXD039572.

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
