# Peer review of "SLC38A10 Deficiency in Mice Affects Plasma Levels of Threonine and Histidine in Males but Not in Females: A Preliminary Characterization Study of SLC38A10−/− Mice"

_genes, 2023, doi:10.3390/genes14040835_

Round 1
Reviewer 1 Report
Dear Authors,
Thank you for submitting the manuscript "SLC38A10 deficiency in mice affects plasma levels of threonine and histidine in males but not in females: a preliminary characterization study of SLC38A10-/- mice". After reviewing the manuscript, various shortcomings have been observed, the major ones are outlined in the attached file.

Author Response
Dear Reviewer 1,
Please see the attachment of our response.
Thank you again for your time.
Kind regards,
Frida Lindberg.

Reviewer 2 Report
Dear Authors, The manuscript is nicely written and well presented. However, the following minor corrections need to be done before this can be recommended for acceptance.
Line 25: It would be better to include keywords not included in the main title.
Line 31: Better not to start a sentence with an abbreviation and check throughout.
Line 34: Provide a reference.
Line 39: Check for correct spaces between words here and elsewhere.
Line 43: Provide the full scientific name when you mention it for the first time.
Line 47: Define the abbreviation when it appears for the first time here and elsewhere.
Line 83: Provide the composition of the diet in detail.
Line 115: Provide the specifications for blood collection tubes and needles.
Line 122: Please provide CV% values for the ELISA tests that you performed.
Line 141: Please follow the text's proper order of figure citations. For instance, after Figure 1A, it should be Figure 1B or 2 or 2A, not 4B. Check the author's guidelines and correct them accordingly.
Line 224: Check for the correct way of writing units after a measurement here and elsewhere. For instance, -80°C or -80 °C.
Line 224: I can see different measurements were taken at various ages of mice. Any particular reason behind that? Please explain.
Line 257: How did you decide the number of animals killed for each analysis? An arbitrary or specific method followed.
Line 437: If my understanding is correct, a table or figure should appear immediately after being cited within the text. Please check and correct accordingly.
Line 641: I am not a person who would like to see citing references in the conclusion section as it is purely dedicated to your overall conclusions. Think about it, and check whether the journal allows such a step.
Author Response
Dear Reviewer 2,
Please see the attachment of our response.
Thank you for your time.
Kind regards,
Frida Lindberg.

Round 2
Reviewer 1 Report
Thank you for revising the manuscript as per the comments and concerns raised. I could see that many of the concerns identified previously have been addressed appropriately, especially those which required more explanation in the text. However, serious concern still remains with regard to the following findings in the manuscript:
1. RNA sequencing experiment. The authors state that RNA sequencing was carried to observe a compensatory mechanism as a result of Slc38a10 KO which did not turn out to be the case here. There is no concrete observation that this experiment supports.
2. Relative telomere length measurement. This experiment has been performed solely on male mice. No genotypic effect was noted in this experiment and therefore this also does not support any conclusions presented in the manuscript.
3. Proteomic measurement in brain. Again, this experiment has been performed solely on male mice. No genotypic effect was noted in this experiment and therefore this also does not support any conclusions presented in the manuscript.
4. 1, 2 and 3 not been carried out on female mice. Although the authors describe that the reason for this exclusion was lack of any noticeable effect on the male mice. However, a possibility of observing an effect with female mice still remains and can not be ruled out unless both these experiments are performed on female mice as well.
Additionally, since the very title of the manuscript is rests on the premise that SLC38A10 deficiency affects the plasma levels of threonine and histidine in male but not female mice, it is important to include female mice in all the experiments performed to rule out any effects on the latter.
5. Since 1,2 and 3 does not yield any noticeable effects that the authors anticipated, it would be appropriate to perform more relevant experiments to clearly demonstrate that the deficiency of SLC38A10 does indeed affect the plasma levels of threonine and histidine in male but not female mice. The current form of the manuscript is interesting but not sufficient to support the conclusions that have been made.
Author Response
Dear Reviewer 1,
Thank you again for your time and valid points made. We have tried to incorporate them into our manuscript, as well as trying to answer them in the attached PDF-file.
Kind regards,
Frida Lindberg.
